# Breeding and Genomics Interventions for Developing Ascochyta Blight Resistant Grain Legumes

**DOI:** 10.3390/ijms23042217

**Published:** 2022-02-17

**Authors:** Uday C. Jha, Kamal Dev Sharma, Harsh Nayyar, Swarup K. Parida, Kadambot H. M. Siddique

**Affiliations:** 1Indian Institute of Pulses Research, Kanpur 208024, India; 2Department of Agricultural Biotechnology, CSK Himachal Pradesh Agricultural University, Palampur 176062, India; kml1967@rediffmail.com; 3Department of Botany, Panjab University, Chandigarh 0172, India; harshnayyar@hotmail.com; 4National Institute of Plant Genome Research (NIPGR), New Delhi 110001, India; swarup@nipgr.ac.in; 5The UWA Institute of Agriculture, The University of Western Australia, Perth, WA 6001, Australia

**Keywords:** grain legume, Ascochyta blight, genomics, molecular marker, QTL

## Abstract

Grain legumes are a key food source for ensuring global food security and sustaining agriculture. However, grain legume production is challenged by growing disease incidence due to global climate change. Ascochyta blight (AB) is a major disease, causing substantial yield losses in grain legumes worldwide. Harnessing the untapped reserve of global grain legume germplasm, landraces, and crop wild relatives (CWRs) could help minimize yield losses caused by AB infection in grain legumes. Several genetic determinants controlling AB resistance in various grain legumes have been identified following classical genetic and conventional breeding approaches. However, the advent of molecular markers, biparental quantitative trait loci (QTL) mapping, genome-wide association studies, genomic resources developed from various genome sequence assemblies, and whole-genome resequencing of global germplasm has revealed AB-resistant gene(s)/QTL/genomic regions/haplotypes on various linkage groups. These genomics resources allow plant breeders to embrace genomics-assisted selection for developing/transferring AB-resistant genomic regions to elite cultivars with great precision. Likewise, advances in functional genomics, especially transcriptomics and proteomics, have assisted in discovering possible candidate gene(s) and proteins and the underlying molecular mechanisms of AB resistance in various grain legumes. We discuss how emerging cutting-edge next-generation breeding tools, such as rapid generation advancement, field-based high-throughput phenotyping tools, genomic selection, and CRISPR/Cas9, could be used for fast-tracking AB-resistant grain legumes to meet the increasing demand for grain legume-based protein diets and thus ensuring global food security.

## 1. Introduction

Grain legumes are a cheap source of plant-based dietary protein and vital micronutrients and vitamins for the human population, lowering the risk of heart-related diseases and type 2 diabetes, and thus playing a crucial role in reducing global food insecurity and eradicating malnutrition-related problems [1,2,3]. However, various biotic stresses, including Ascochyta blight (AB), a fungal foliar disease belonging to the class Dothideomycetes, order Pleosporales, and family Didymellaceae [4], cause significant yield losses in various grain legumes across the globe [5,6,7,8,9,10,11]. The blight disease in legumes is caused by the fungal species of genus *Didymella* (anamorph stage: *Ascochyta*) belonging to the class Dothideomycetes, order Pleosporales, and family Didymellaceae [4]. Furthermore, various virulent pathotypes or races of AB are becoming a serious concern for sustaining global grain legume yield. AB is a soil-borne disease, and AB pathogens perpetuate on plant inoculum and infected debris of the previous crop [8]. The AB pathogens invade the host and stomata by their appressoria or penetration peg [12,13,14]. Thereafter, the AB pathogen secretes carbohydrate active enzymes, such as glycoside hydrolases, glycosyl transferases, and secretome, that degrade cell celluloses, hemicelluloses, and chitin, spreading the infection inside host plant cells and ultimately causing cell death [15,16]. In response to AB attack, host plants well equipped with sophisticated immune system recruit a two-tier defense system, involving PAMP-triggered immunity (PTI) [17] and effector-triggered immunity (ETI) [18] to restrict entry of the invading AB pathogen. Several approaches have been used to minimize AB-infection losses, including developing AB-resistant cultivars using conventional and modern breeding tools. By relying on Mendelian genetics, several genetic determinants conferring AB resistance have been reported in various legumes [19,20,21,22,23,24,25,26,27]. Since, AB resistance in several crops is polygenic in nature, the advent of molecular marker technology has enabled the identification and location of quantitative trait loci (QTL) conferring AB resistance in grain legume crops using biparental mapping and genome-wide association studies (GWAS). Subsequent advances in next-generation sequencing enabled the construction of the genome assemblies of various grain legumes and the AB-causal organism and thus the exploration of AB-resistant genomic regions in plants and pathogenicity/virulence regions in the AB pathogens. In parallel, whole-genome resequencing (WGRS) and pan-genome assembly have underpinned structural genomic regions conferring AB resistance at the whole-genome level and across plant species. Likewise, the advent of RNA-seq has assisted in uncovering AB-resistant candidate gene(s) and their plausible functions. Proteomics has been instrumental in unveiling various candidate proteins conferring AB resistance and various AB proteins mediating pathogenicity in the host plant. Emerging innovative breeding approaches, such as high-throughput phenotyping, genomic selection, rapid generation advancement, and CRISPR/Cas9 based genome editing technology, will enhance the development of AB-resistant climate-resilient grain legumes for sustaining global legume yields. 

## 2. Causal Organism of AB in Various Legumes, Symptoms, and Negative Impact

### 2.1. Causal Organism of AB in Chickpea, Symptoms, and Negative Impact

AB disease in chickpea is caused by *Ascochyta rabiei* (Pass.) Labr. (teleomorph: *Didymella rabiei* (Kovacheski)), a necrotopic fungus belonging to the class Dothideomycetes, order Pleosporales, and family Didymellaceae [4]. AB can cause up to 100% yield loss in chickpea under severe infection [6,10]. According to Murray and Brennan [28], AB infection causes chickpea yield losses of $4.8 million annually in Australia. Based on AB disease severity, existence of three pathotypes [29,30,31,32], five pathotypes [6], and 10 pathotypes [33] have been reported (see Table 1). Aggressiveness of AB-causing pathotypes may greatly change depending on geographical distribution and pedoclimatic conditions [31]. 

AB is a seed-borne disease, with infected chickpea seed acting as the source of primary inoculum for disease infection [8]. AB completes its sexual cycle on infected chickpea debris over winter [49]; the ascospores produced on the crop residue persist on the soil surface over winter [5,50,51,52]. After disease establishment in the field, asexual spores (pycniospores) serve as an important source of secondary spread of AB [52]. 

### 2.2. Causal Organism of AB in Faba Bean, Symptoms, and Negative Impact

AB disease in faba bean is caused by *Ascochyta fabae* Speg. (teleomorph *Didymella fabae*), significantly reducing yield [53] by up to 90% in susceptible cultivars under wet and congenial conditions [7,54]. The existence of the physiological race of *Ascochyta fabae* has been reported [7,35]. Yield losses ranging from 35 and 90% due to infection caused by *A. fabae* have been reported in winter and early spring grown crops in Middle East, Europe, Canada, and Australia [55,56]. The disease is prevalent in crops grown during winter, spreading widely during spring [55,57] (see Table 1). As AB is a seed-borne disease, infected seeds and crop debris serve as the main inoculum for disease infection on leaves, stems, and pods, resulting in lodging, stem girdling, and pod and seed abortion in faba bean [36]. Multiple environment testing for AB resistance revealed high G×E interactions for AB severity [58]. 

### 2.3. Causal Organism of AB in Pea, Symptoms, and Negative Impact

AB in pea, also known as ‘black spot’ disease, is caused by a complex of three pathogens *Mycosphaerella pinodes* (the teleomorph of *Didymella pinodes*), *Ascochyta pisi*, and *Phoma medicaginis* var. *pinodella* [41,59,60]. The disease is prevalent in all pea-growing regions, including Europe, Mediterranean basin, North America, and Australia, causing significant yield losses [8,9,11,45,61]. For example, annual yield losses caused by AB in pea are 10–60% in Australia [45], 40% in France [62], 50% in Canada [63], and 10–30% in China [64]. AB infection starts at the seedling stage and spreads across aerial parts; visual symptoms include necrotic leaf spots, chlorotic halos in leaves, stem and pod lesions, and dark brown discoloration of seeds [44,64,65,66] (see Table 1). 

### 2.4. Causal Organism of AB in Lentil, Symptoms, and Negative Impact

AB in lentil is caused by *Ascochyta lentis* Vassiljevsky (teleomorph: *Didymella lentis*, syn. *Ascochyta fabae* f. sp. *lentis*) [37]. This disease is found in all major lentil-producing countries, including Canada, Australia, India, and New Zealand [67,68]. *Ascochyta lentis* is host-specific, so it does not show any disease symptoms on host species other than lentil [37]. AB disease symptoms appear as necrotic lesions on leaves, stems, and pods, inhibiting photosynthesis and causing up to 70% seed yield losses [5]. Infected seed or pycniospores left in soil from previous lentil crops act as the main inoculum for onset of disease infection [8,9,69]. 

### 2.5. Causal Organism of AB in Grasspea, Symptoms, and Negative Impact

AB in lathyrus, caused by *A. lentis* var. *lathyri*, significantly reduces seed yield. The disease symptoms are characterized by necrotic lesions on the stems and leaves [39].

## 3. Ascochyta Blight Infection and Underlying Host Plant Resistance Mechanism

AB pathogens penetrate the first layer of the plant defense system conferred by host cuticle and epidermal cells using penetration pegs [12,13,14,15,70]. The pathogen spores land on the surface of the host and germinate to form germ tubes that bear appressoria and penetration pegs. The pathogens also secrete carbohydrate active enzymes, including glycoside hydrolases, glycosyl transferases, and carbohydrate esterases, to degrade cellulose, hemicellulose, and chitin and spread infection inside host plant cells [15,16]. Following penetration, the pathogen establishes organic relationships with the host cell, hyphae multiply and spread infection in the host. In retaliation to pathogen attack, the host generates oxidative stress to kill or damage the pathogen hyphae. At least one AB pathogen, *A. rabiei*, possesses genes to overcome oxidative stress and survive under oxidative stress generated by the host during the pathogen invasion. A transcriptome analysis of *A. rabiei* under oxidative stress and control indicated involvement of genes *viz.*, *ST47_g10291*, *ST47_g9396*, *ST47_g10294*, *ST47_g4395*, and *ST47_g7191* surviving under the oxidative stress and enhancing pathogenicity [71]. To establish successful infection, AB pathogens have to overcome the host’s PTI- and ETI-mediated defense mechanisms.

In response to AB attack, the host plant recruits two lines of defense (1) PTI, or basal defense mechanism [17,72] and (2) ETI [18]. During PTI, the host plant recognizes microbial elicitors/PAMPs [73] using pattern recognition receptors embedded in the cell membrane, activating the plant defense mechanism [73] (see Figure 1). After perceiving the PAMP signal, the pattern recognition receptors (e.g., receptor-like protein kinases [80,81] and brassinosteroid insensitive 1-associated kinase 1 [82] induce reactive oxygen species that enhance the influx of cytosolic Ca^2+^ and trigger mitogen-activated protein kinase signal cascades [74]. Subsequently, various host phytohormones (e.g., jasmonate, ethylene, ABA) are induced to activate various downstream target transcription factors (TFs), such as WRKY and ERF, and switch on downstream transcriptionally active genes encoding defense-related proteins (e.g., pathogenesis-related proteins, osmotins providing plant protection against the invading AB pathogens) [75,76,77,78]. During ETI—the second tier of defense mediated by host disease resistance(R) gene(s)—the effector molecules secreted by the AB pathogen are recognized by host plasma membrane-based nucleotide-binding/leucine-rich repeat (NLR) receptors with CC-domain NLRs (CNL) and HeLo-domain RNLs that form Ca^2+^-permeable channels [18,79] (see Figure 1). The subsequent enhanced influx of cytosolic Ca^2+^ renders enhanced pattern recognition receptor immunity mediating pathogen and host cell death due to a hypersensitive response [79]. PTI and ETI interconnect through signaling networks and downstream responses [80].

To avert ascochyta blight (AB) pathogen intrusion, host plant employs two types of defenses. First tier basal defense mechanism is pathogen-associated molecular pattern (PAMP) triggered immunity (PTI). Pattern recognition receptors embedded in host cell (cell membrane) sense the PAMP molecules of the pathogen [17] followed by activation of complex circuit of signal transduction involving reactive oxygen species (ROS), mitogen activated protein kinases, Ca^+2^ [74] and phytohormones viz., abscisic acid, salicylic acid and jasmonate [75,76,77]. The signal molecules activate transcription factor (ERF and WRKY) genes [75,76,77] leading to generation of transcription factors that eventually switch on the downstream gene(s) encoding chitin, osmotin and pathogenesis related proteins that restrict the AB infection [78]. If PTI mechanism of host defense fails to restrict AB pathogen attack, plant deploys second tier of defense system known as effector triggered immunity (ETI) [18]. In this mechanism host plant senses the effector molecules secreted by attacking AB pathogen [39] through intracellular nucleotide-binding/leucine-rich repeat (NLR) receptors and activate host plant resistance (R) genes encoding nucleotide-binding/leucine-rich repeats that ultimately lead to hypersensitive response and host cell death [75,79]. Both PTI and ETI mechanisms are interconnected with signaling networks and downstream responses [80]. ETI in association with Ca^+2^ signaling boosts host PTI [77,79]. 

## 4. Genetics of AB Resistance

Classical genetic studies have revealed the genetics of AB resistance in various grain legumes, evidenced by monogenic inheritance controlled by a single dominant gene [19,20,23,83], single recessive gene [21,22,23,24,84], or digenic inheritance [25,26,27]. There are also reports of quantitative inheritance for AB resistance [27,85].

The inheritance pattern of AB resistance in chickpea I-13 variety is controlled by a single dominant gene based on an analysis of three cross combinations and their reciprocal crosses by assessing them in an Aschochyta infected sick plot [86]. Likewise, a preliminary screening of F_1_ and F_2_ populations of crosses between five resistant and 11 susceptible parents, [84] advocated that AB resistance is inherited as a single dominant gene in ILC 72, ILC 183, ILC 200, and ILC 4935 and a single recessive gene in ILC191. The authors also proposed *rar*1 as a gene symbol for recessive genes and *rar*2 for dominant genes controlling AB resistance in chickpea. Screening F_2_ and F_3_ populations derived from crosses of resistant (ILC 72, ILC 202, ILC 2956, and ILC 3279) and susceptible chickpea parents in epiphytotic condition revealed that resistance against Ascochyta blight causing race 3 is governed by a single dominant gene [19]. An allelic test confirmed that the resistant gene in the four resistant parents is the same [19]. Similarly, Tewari and Pandey [87] reported that genetic inheritance of AB resistance is controlled by a single dominant gene after screening F_2_, BC_1_, BC_2_, and F_3_ crosses between six resistant and four susceptible parents under field and glasshouse conditions. The allelic test confirmed the presence of three independent genes controlling AB resistance, one dominant gene in P 1215-1 and another in EC 26,446; and PG 82-1; and one recessive gene in BRG 8 [87]. Dey and Singh [25] proposed two dominant complementary genes from the GLG 84,038 and GL 84,099 genotypes and one dominant and one independent recessive gene from the ICC1468 genotype. The identified genes were symbolized as Arc_1_, Arc_2_ (GLG 84038), Arc_3_, Arc_4_ (GL 84099), and Arc_5(3,4)_ (ICC1468). A generation mean analysis confirmed that additive gene action conferred the resistance by GLG 84,038 and GL 84099, and dominance and dominance × dominance interaction was present in ICC1468 [25]. Danehloueipour et al. [88] scored the disease resistance reaction of 5 × 5 half-diallel chickpea crosses under field conditions to reveal quantitative control (additive and dominance gene action) of AB resistance. 

In lentil, studies have shown that genetic control of AB resistance is governed by a single dominant gene [20,21,23,26,83,89,90], single recessive gene [21,22,23,24], two dominant genes [22], two dominant complementary genes [26,91], or two additive recessive genes, and two duplicated recessive genes [23,89]. 

In faba bean, one study reported that genetic inheritance of AB resistance is controlled by a major dominant gene in ILB752 and minor genes in NEB463 [92]. Earlier, Maurin and Tivoli [93] reported AB resistance in a 29H genotype based on its low disease index scores tested for two consecutive years (1985–1986 and 1986–1987) in the field. Maurin et al. [94] also reported AB resistance in a 29H genotype by recording its hypersensitive reaction in a histopathological test resulting in flecking lesions on the host plant.

An AB resistance study in pea found *Rmp1* and *Rmp2* genes controlling stem resistance and *Rmp3* and *Rmp4* genes controlling leaf resistance based on the segregating pattern of crosses between resistant lines (JI 97 and JI 1089) and a susceptible line [13]. Subsequently, Rastogi and Saini [27] reported that AB resistance is governed by two independent dominant genes using segregating population (F_1_, F_2_, and F_3_) data from Kinnauri (resistant) crossed with Bonneville, Lincoln, GC 141, and Selection 18 (susceptible) genotypes. However, Wroth [95] reported quantitative inheritance of AB-disease response in nine pea genotypes using diallel analysis. Other studies have also reported additive and dominant effects for conferring AB resistance in pea [85,96]. 

## 5. Legume Crop Diversity and Genetic Resource: Economic and Sustainable Approach for Developing AB Resistance

Among the various approaches for controlling AB infection and minimizing yield losses caused by AB, using genetic resources in cultivated and crop wild relatives (CWRs) remain the most effective, environmentally friendly, resource-saving, and economically profitable approach for developing AB-resistant legumes [13,97,98,99,100]. An abundance of genetic variability for AB resistance has been reported in cultivated chickpea [101,102,103,104,105]. The ILC482 genotype, with slow blighting and partial AB resistance, was released in eight countries [101]. An evaluation of 36 chickpea lines at two locations in Kenya identified ICC7052, ICC4463, ICC4363, ICC2884, and ICC7150 as AB resistant under field conditions [102] (see Table 2). A multi-environment screening identified IC275447, IC117744, EC267301, IC248147, and EC220109 with AB resistance under field conditions [105].

Unlike cultigen, CWRs of legume crops are rich in allelic diversity for resistance to various biotic and abiotic stress tolerances, including AB resistance in legume crops [99,106,114]. Various studies have identified CWR potential donors for AB resistance in chickpea: *C. echinospermum* [122], *C. reticulatum* [99], and *C. judaicum* and *C. pinnatifidum* [106,129]. A screening of 201 accessions of eight annual wild Cicer species identified *C. judaicum* and *C. pinnatifidum* accessions as sources of AB resistance [106]. Another study identified *C. echinospermum* and *C. reticulatum* accessions as sources of AB resistance [99]. Newman et al. [114] reported AB resistance in *C. echinospermum* and *C. reticulatum* accessions collected from southeastern Turkey.

Lentil has a substantial amount of genetic variability for AB resistance. Several donors of cultivated species confer AB resistance in lentil, including Indian head [89], Laird [21], ILL5588 [20,24,83,130], and ILL5684 [21,22]. Targeting novel sources for AB resistance in lentil, a focused identification of germplasm strategy assisted in identifying 87 landraces, including IC207, as AB resistant after assessing 4567 accessions against *A. lentis* FT13037 isolate in the field [124]. Likewise, potential of lentil CWRs conferring AB resistance has been reported in lentil [98,131]. 

Among the various lentil CWRs, *Lentis orientalis*, *Lentis odemensis*, *Lentis nigricans*, and *Lentis ervoides* accessions are promising sources of AB resistance in lentil [123,126,131]. Accessions W63241 and W63261 (*L. orientalis* [23,26,83,90], W63192 (*L. ervoides*; [26,132], W63222 (*L. odemensis*; [26], and ILWL235 (*L. odemensis*; [133] have been used to determine the genetics of AB resistance and hence could be used as donor parents for developing AB-resistant lentil cultivars.

Infection caused by *A. fabae* reduced faba bean yields by 35–90% in winter and early spring grown crops in the Middle East, Europe, Canada, and Australia [55,56]. In 1975 in Cambridgeshire, a field study on AB infection in faba bean identified several tolerant genotypes (IB19, Bulldog, Banner, Buccaneer, IB7CS, IB18, and Maris Beagle) [53]. A thorough screening of 672 faba bean accessions across Syria, England, Canada, Poland, France, and Tunisia over three years (1983–1985) identified BPL 471, 460, 646, 74, and 2485 genotypes as AB resistant across all the tested locations [7]. Likewise, a field assessment of 752 faba bean germplasm identified 34 lines with low severity against AB that could be used to develop AB-resistant faba bean varieties [118]. Multi-environment testing of 484 faba bean accessions across two seasons in the Czech Republic, Estonia, Germany, and Spain revealed L-831818, V-26, and V-958 as the most stable and resistant across all locations [58].

Several sources of AB resistance in pea have also been identified. Kraft et al. [61] examined 2936 pea accessions for AB resistance in the year 1991, 1992, and 1994; 157 accessions were further tested for AB resistance in 1995 at Carlow in Ireland and Gore in New Zealand. Five genotypes (PI 142441, PI 142442, PI 381132, PI 404221, and PI 413691) were identified as AB resistant. Field screening of 500 pea lines tested for AB resistance in Ethiopia in 1998 identified 40 lines with partial resistance [134]. A thorough assessment of 335 pea lines during the years 1994 and 1995 in an AB-inoculated field revealed seven pea genotypes with AB tolerance [135]. Further, these lines were tested for partial resistance against AB during the years 1996–1998, of which, Baccara and Yellowhead had small yield reductions of 10 and 17%, respectively, under AB infection [135]. Based on a detached leaf assay for AB resistance, Zhang et al. [136] recorded substantial genetic variability for AB resistance in 558 pea genotypes tested in the field for two years. Wild relatives of pea are also important sources of resistance to AB. A high degree of resistance against AB was reported in wild *Pisum* species, *P*. *fulvum*, *P. sativum* ssp. *elatius* and *P. sativum* ssp. *syriacum* [13,100,127] (see Table 2). These CWRs could serve as important donors of AB resistance for development of prebreeding material for transfer of resistance to elite cultivars of pea.

## 6. Identification of AB-Resistant QTL Using Biparental Mapping and Genome-Wide Association Studies

Advances in molecular marker technologies including development of several types of markers such as random amplified polymorphic DNA (RAPD), sequence-tagged sites (STSs), simple sequence repeats (SSRs), and single nucleotide polymorphisms (SNP), have led to the identification of QTLs/genomic regions governing AB resistance by using a biparental mapping approach in various legumes [128,137,138,139,140,141,142,143].

Using a biparental mapping approach, incorporating cultivated and wild cross Lasseter × *C. echinospermum*, Collard et al. [144] mapped one QTL contributing to AB resistance on LG4. Likewise, Cobos et al. [140] mapped one QTL on LG2 from a *Cicer arietinum* (ILC72) × *Cicer reticulatum* (Cr5-10) cross using RAPD, ISSR, STMS, and isozyme markers. Three QTL (*ar1*, *ar2a*, *ar2b*) were mapped on LG2 and LG4 [145] whereas three AB resistant QTLs, two on LG4 and one on LG8 were identified by genotyping of RIL derived from kabuli × desi cross with SSR markers [146] (see Table 3). Irulea et al. [149] mapped one QTL *QTL_AR3_* on LG2. This *QTL_AR3_* was further fine mapped on Ca2 with physical position 32–33 Mb, containing 42 candidate genes including genes *Ein3*, *Avr9/Cf9* and *Argonaute 4* genes participating in disease resistance mechanism [156]. Likewise, EIN4-like sequence (*CaETR-1*) was uncovered in QTL(AR1) (obtained from WR315 × ILC3279 mapping population) on LGIVa flanked by NCPGR91 and GAA47 SSR marker explaining 33.8%PV [139]. Five major QTLs explaining 14–56% phenotypic variation on LG2, LG3, LG4, LG6, and LG8 were discovered from four segregating mapping populations [151]. Furthermore, three QTLs were reported on LG3 and LG4, explaining 49% PV [153]. An SNP marker in chickpea was used to discover *qABR4.1*, *qABR4.2*, and *qABR4.3* QTLs and a *CaAHL18* candidate gene on Ca4 explaining 42% PV using two recombinant inbred lines developed from interspecific and intraspecific crosses using multiple quantitative trait loci sequencing (mQTL-seq) [142]. The authors further narrowed the *qABR4.1* genomic region from 4.476 to 4.675 Mb (~200 kb), flanked by CaNIP18 and CaNIP12 markers on Ca4, corresponding to previously identified AB resistant QTLs *QTLAR1* and *QTL1* [139,181] and suggesting its conservation across different chickpea genotypes. A next generation sequencing-based bulk segregant analysis (BSA) in two populations, CPR-01 and CPR-02, recovered 11 AB-resistant QTLs from CPR-01 and six AB-resistant QTLs from CPR-02 on Ca1, Ca2, Ca4, Ca6, and Ca7, explaining 13–19% PV [162]. Notably, among these QTLs, QTL *CPR01**-qAB1.1* showed overlap with the AB-resistant QTL on the same genomic region previously reported by Daba et al. (2016) [158] from a CPR-01 population using a conventional mapping approach.

Deokar et al. [161] uncovered eight QTLs conferring AB resistance by genotyping RIL developed from Amit × ICCV96029 using Illumina^®^ GoldenGate array on LG2, 3, 4, 5, and 6, explaining up to 70% PV. Recently, two *Cicer echinospernum* QTLs *AB_echino_2014* and *AB_echino_2015*, for AB resistance were reported on LG4 using interspecific mapping populations derived from *C. arietinum* × *Cicer echinospernum* [115]. Two AB-resistant QTLs, *qab-4.2* on LG4 explaining 10.6% PV and *qab-7.1* on LG7 explaining 8.2% PV were detected consistently in the same genomic region of a GPF2 × ILWC292 mapping population screened over two consecutive years [163]. 

Similar to chickpea, several studies were conducted to identify AB-resistant QTLs in lentil [20,120,170,171,172]. Using RAPD, AFLP, and ISSR markers, Tar’an et al. [171] mapped two AB-resistant QTL on LG4. Similarly, six QTL explaining up to 69% PV were reported on LG1, 2, 4, and 5 [120]. QTL-5 on LG1 and QTL-3 on LG4 obtained by Rubeena et al. [120] overlapped QTL-1 and QTL-5, respectively, reported by Gupta et al. [173]. Another AB-resistant QTL, mapped on LG6, explained 41% PV [172]. Screening a population derived from Indianhead × Northfield using SNP and SSR markers identified four resistant QTLs, *AB_IH1*, *AB_IH1.2*, *AB_NF1*, and *AB_IH1.3* QTL on LG2, 3, and 6 [138] (see Table 3). Of these four QTL, *AB_NF1* shared a common genomic region and AB-resistant candidate gene with three major QTL on LG6, explaining up to 27% PV, from an interspecific cross of *L. culinaris* × *L odomensis* [133]. Subsequent genotyping of an ILWL 180 × ILL6002 population using GBS-derived SNP markers uncovered one major QTL along with three minor QTL for AB resistance on LG5 explaining 9.5–11.5%PV, and one QTL on LG2 explaining 9.6% PV [126]. Notable candidate genes underlying the QTL, including cinnamoyl-CoA reductase 1, phenylalanine-tRNA ligase, and ferredoxin-dependent glutamate synthase were uncovered [126]. 

Using RAPD, STS, STMS, and CAPS markers, used in a ATC80878 × ATC 80,407 backcross mapping population of faba bean, identified two QTL (QTL1 and QTL2) contributing to AB resistance on LG1 and LG2, explaining 9–12% PV [169]. Screening of an F_2_-based mapping population in faba bean identified six AB-resistant QTL, with *Af3* and *Af4* showing resistance against CO99-01 and LO98-01 isolates of AB, while, *Af5* exhibiting resistance against isolate CO99-01, and *Af6*, *Af7*, and *Af8* showing resistance against isolate LO98-01 [165].

Likewise, based on disease severity on stems and leaves, two putative QTL (*Af1* and *Af2*) on LG2 and LG3 were reported in the Vf6 × Vf136 RIL-based mapping population [166]. Atienza et al. [57] confirmed *Af2* on chromosome 2, but *Af3* on chromosome 3 did not coincide with *Af1* reported by other researchers investigating AB-resistant QTL in the 29H × Vf136 mapping population. Gutierrez and Torres [143] identified three *Af2* QTL on chromosome 2, two *Af3* QTL on chromosome 3, and *F_DSP1*, *F_DSP2* and *DSL_Lo98* three QTL on chromosome VI, with *Af2* consistently observed in field and growth chamber experiments over three years. There were 748 underlying candidate genes were predicted in the *Af2* QTL interval [56] (see Table 3), of which *Medtr3g099380* encoding 14-3-3 like protein and *Vf_Medtr3g099010* encoding HVA22-like protein were involved in conferring disease resistance [56]. Likewise, QTLs obtained on chromosome VI by Gutierrez and Torres [143] coincided with QTL reported by Ocaña-Moral et al. [56]. The possible candidate genes underlying these QTL were *Medtr8g095030* and *Medtr4g087620* encoding leucine-rich repeat receptor-like serine/threonine protein kinase [143]. In a study to validate these QTL from 29H × Vf136 mapping, Gutierrez and Torres [143] recorded two *Af2* on chromosome 2 and two *Af1* QTL on chromosome 3 for disease resistance in stems and leaves in the Vf6 × Vf136 population. These QTL remained in the same regions reported by Díaz-Ruiz et al. [166]. Gutierrez and Torres [143] also reported that an important candidate gene *Vf_Medtr3g102180* underlying *Af*2 QTL encoded inactive receptor kinase mediating plant immunity in response to disease and *Medtr1g106005* gene underlying *Af*1_DSS QTL encoded α-tubulin contributing to stress response signaling.

Several QTLs associated with AB resistance were also identified in pea, however, the number of QTLs was higher than that in other legume crops included in this study. In a study mapping AB-resistant QTL in pea, Timmerman-Vaughan et al. [174] mapped 13 QTL on seven linkage maps for AB resistance from a F_2:3_ population developed from 3148-A88 × Rovar. Of these QTL, eight were identified under multiple environments. Similarly, phenotypic evaluation of RILs developed from Carnival × MP1401 cross and genotyping with RAPD, AFLP, and STS markers identified three AB-resistant QTL on LGII, IV, and VI, explaining 36% PV [154] (see Table 3). QTL on LGIV shared the same region for AB resistance reported by Timmerman-Vaughan et al. [174]. Further phenotypic screening and genotyping of two mapping populations developed from A26 × Rovar and A88 × Rovar identified 11 and 14 QTL on all LG groups explaining 4.6–37.4% PV [176]. Of the QTL identified from these two populations, six shared a common genomic region for AB resistance. The QTL identified from A26 × Rovar coincided on the same genomic region as QTL *Asc2.1*, *Asc3.1*, *Asc5.1*, and *Asc7.1* in the A88 × Rovar population reported by Timmerman-Vaughan et al. [174]. Aiming at mapping QTL for partial AB resistance at seedling and adult plant stage, a total of six QTLs on LGII, Va, VI, and VII were identified on stipules and stems at the seedling stage under controlled conditions explaining upto 20% phenotypic variation from the JI296 × DP, RIL mapping population [137]. From the same study another 10 QTLs on LGII, III, Va, and VII were identified on stipules, stems, or for both organ in adult plants under field conditions explaining 6–42% PV [137]. The QTL on LGIII coincided with the AB-resistant QTL *Asc3.1* already reported by Timmerman-Vaughan et al. [176]. Likewise, Fondevilla et al. [175] reported six QTL on LGII, LGIII, LGIV, and LGV, explaining 31–75% PV. Furthermore, three new AB-resistant QTL on LGIII and LGVI were uncovered [178]. Considering various plausible AB-resistant candidate genes, Prioul-Gervais et al. [177] underpinned candidate genes *PsDof1* and *DRR230-b* coinciding with QTL *mpIII-1* and *mpIII-4* on LGIII reported by Prioul et al. [137]. Moreover, this genomic region on LGIII controlling AB resistance was colocalized with QTL *Asc3.1* [174] and *QTL MpIII.1* [175] governing AB resistance. Another colocalization of the AB-resistant genomic region occurred on LGVII, where Prioul-Gervais et al. [177] found some important resistance gene analogs (*RGA2*, *RGA3*, *RGA-G3A*, *IJB174*, and *IJB91*) coinciding with QTL *mpVII-1* [137]. In this context, Jha et al. [182] found significant SNPs within *PsDof1* (PsDof1p308) and *RGA-G3A* (RGA-G3Ap103) candidate genes. Of the four new AB-resistant QTL (*MpII.1*, *MpIII.5*, *MpV.2*, and *MpV.3*) identified by Carrillo et al. [183], *MpIII.5* coincided with QTL *mpIII.2* reported by Prioul et al. [137]. Moreover, the authors unveiled candidate gene *ArfB3*, residing into *MpV.1_DRseedl* (encoding *auxin response factor B3 domain*), and *CE007J22* (encoding *hypersensitive-induced reaction protein 4*) coinciding with QTL *MpVI.1*. Subsequently, Jha et al. [128] mapped nine AB-resistant QTL explaining 7.5–28% PV from a P651 (*P*. *fulvum*) and Alfetta (*Pisum sativum* L.) interspecific RIL-based mapping population. Of these QTL, two *abIII-1* and *abI-IV-2* were consistent across the tested locations and years and later fine mapped using GBS-derived SNP markers by genotyping heterogeneous inbred family (HIF)-224 and HIF-173 derived from F_6_ RILs of PR-19-224 and PR-19-173 [141]. Two new QTL, *abI-IV-2.1* and *abI-IV-2.2*, explaining 5.5–14% PV, were discovered within QTL *abI-IV-2*. Ten of 12 SNPs derived from GBS were mapped within QTL *abI-IV-2*; three markers (Sc34405_60551, Sc33468_44352, and Sc12023_67096) resided within the nearest markers PsC6805p316 and PsC19558p107 [141]. Numerous AB-resistant QTL have been reported in various legumes; however, few have been cloned to decipher specific candidate gene(s) and their function controlling AB resistance [142]. Thus, future studies should focus on fine mapping/cloning the identified AB-resistant QTL to underpin candidate genes conferring AB resistance in grain legumes.

## 7. From Marker Assisted Selection to Genomic Selection Developing AB-Resistant Grain Legumes

Practice of marker assisted selection is primarily successful for transferring of major gene(s)/major QTL/large effect QTL [184]. Introduction of quantitative resistance genes/minor genes through MAS is challenging because of high impact of genotype × environment effect on these minor gene(s)/QTLs [185,186]. Abundance of high-throughput molecular markers owing to advances in genomic research and availability of genome sequence assembly in various legumes have allowed embracing genomic selection (GS) [187]. Thus for complex traits, to predict genetic merit of selection candidate/to select potential progeny having high resistance to AB in grain legumes without testing under field condition, genomic selection (GS)/genomic prediction could be a new avenue to develop AB resistant grain legume [188]. 

The accuracy of GS for predicting AB resistance in pea based on Ascochyta blight disease score (ASC) was recorded for two consecutive years in a training population of 215 lines using SNP markers derived from GBS—the GBLUP analysis produced the best prediction accuracy for ASC (0.56) [186].

## 8. Genome-Wide Association Mapping for Uncovering AB-Resistant Genomic Regions across the Whole Genome in Grain Legume

Genomewide association study (GWAS) is a powerful genomic approach for uncovering genetic variants across the whole genome in a large panel of global plant population to identify genotype-phenotype association [189]. Substantial efforts have been devoted for investigating AB-resistant genomic determinants in various legumes [114,159,168]. In parallel, high throughput markers developed from whole genome resequencing efforts further facilitated in conducting GWAS for elucidating AB-resistant novel genomics region. Aiming at this, combined efforts of WGRS and GWAS identified an AB-resistant genomic region (QTL *AB4.1* on LG4) and narrowed the region that overlapped the AB-resistant genomic regions obtained from Hadas × ICC5810 [146], C 214 × ILC 3279 [155], and Lasseter × ICC3996 [157] RIL populations from 7 Mb, 13 Mb, and 30 Mb to 100 kb [159]. The study also identified 12 candidate genes, including *Ca_05515* encoding LRR receptor-like kinase, *Ca_05520* encoding wall-associated kinase, *Ca_05511* encoding zinc finger protein, *Ca_05516* encoding cysteine-rich receptor-like kinase, and *Ca_05517*, *Ca_05521*, *Ca_05522*, and *Ca_05523* encoding serine/threonine protein kinases [159]. A GWAS on 149 accessions of *C. reticulatum* using RAD-seq data underpinned four significant candidate genes—*Cr_ 02657.1* encoding WRKY transcription factor on LG1, *Cr_09847.1* encoding auxin response factor on LG4, *Cr_16402.1* encoding sugar transporter on LG6, and *Cr_08467.1* encoding spermatogenesis-associated protein 20 on LG3 [114]. The SNP identified on *C. reticulatum* on chromosome 4 corresponded to *C. arietinum* chromosome 4 that colocalized with the AB-resistant *QTL_AR5_* identified by Deokar et al. [161] and Sabbavarapu et al. [155]. In addition, the SNP identified on chromosome 3 corresponded to *C. arietinum* chromosome 3 that overlapped *QTL_AR4_* reported by Tar’an et al. [150]. Faridi et al. [168] conducted GWAS in 188 inbred faba bean lines using 1829 AFLP and 229 SNP markers and reported 12 significant MTAs (explaining 5.6–21.7% PV) associated with six AB-related traits. One significant SNP (Vf_Mt1g014230_001) on chromosome 3 overlapped a previously reported AB-resistant QTL*Af1* [190].

## 9. Functional Genomics

Several microarrays and DeepsuperSAGE analyses have been undertaken to gain insights into the role of gene(s) contributing to AB resistance in legumes [191,192,193].

A cDNA library constructed from leaf and stem tissue of AB-resistant ATC 80,878 lathyrus genotype infected with AB at 48 and 72 h post-infection revealed 818 AB-responsive ESTs, of which 50 were related to the AB-disease response, and their predicted functions were related to pathogenesis-related proteins, phenylpropanoid pathway, and defense mechanisms against disease [192]. Coram and Pang (2005a) [191] conducted a microarray study in two contrasting chickpea parents—ICC3996 (AB-resistant) and Lasseter (AB susceptible)—to gain insights into the expression of 20 defense-related ESTs, revealing upregulatory action of “leucine zipper protein”, “SNAKIN2 antimicrobial peptide precursor”, and “elicitor-induced receptor protein” genes in ICC3996. Subsequently, Coram and Pang [193] undertook a large-scale gene expression analysis of AB-resistant, susceptible, and moderately resistant chickpea lines and wild species using a microarray study. Of the 756 microarrays, 97 genes were differentially expressed in at least one genotype at one time-point. The genes involved in conferring AB resistance were “pathogenesis-related proteins”, “proline-rich protein”, SNAKIN2 antimicrobial peptide, leucine zipper protein, and Ca-binding protein [193] (see Table 4).

Functional analysis of AB-inoculated and control leaves of grasspea using deepSuperSAGE revealed 14,387 UniTags, of which 738 were differentially expressed between inoculated and control leaves [197]. Defense-related genes revealed upregulatory action controlled by the ethylene pathway [197]. The study also revealed overexpression of cellulose synthase and lignin biosynthesis genes contributing to cell wall metabolism, upregulation of *chitinase A* (*PR-3*) restricting fungal hyphal growth, and upregulation of *glutathione S-transferase*, *phytoene synthase* involved in ROS detoxification [197]. To understand the genic bases of AB resistance in pea, microarray analysis of AB-inoculated resistant (P665) and susceptible (Messire) accessions revealed differential expression of genes such as phenylpropanoid and phytoalexins metabolism, pathogenesis-related (PR) proteins and those involved in jasmonic acid (JA) and ethylene signal transduction in response to pathogen infection [196]. Subsequent validation of ten differentially expressed genes using qRT-PCR revealed enhanced expression of defense-related genes (glutathione S-transferase, peroxidases, and 6a-hydroxymaackiain methyltransferase) in P665 [196] (see Table 4). 

Unprecedented advances in functional genomics fueled by RNA-seq based technologies have elucidated candidate gene(s) with plausible functions related to various biotic stress resistance, including AB resistance in legumes [70,75,76]. RNA-seq analysis of two contrasting lentil genotypes—ILL7537 (resistant) and ILL6002 (susceptible)—in response to AB revealed upregulatory and downregulatory activity of myriad of genes, including pathogen recognition signaling molecules (*LRR receptor-like kinase*, *Calmodulin domain protein kinase-like*), pathogenesis-related and anti-fungal compounds (*polygalacturonase inhibitor*, *superoxide dismutase*, *PR protein 2—O-glycosyl hydrolase*), defense-related transcripts (*ethylene response factor*), and hypersensitive response (*gibberellin signaling DELLA protein*, *gibberellin receptor*) at different time scales post-AB infection [70]. Sari et al. [122] also reported differential expression of various genes in contrasting lentil genotypes subjecting AB infection, such as those involved in pathogen recognition, nucleotide-binding site leucine-rich repeat (NBS-LRR) receptors, phytohormone signaling, pathogenesis-related proteins, cell wall enforcement, and cell death. Similarly, García-García et al. [195] also elucidated the inductive role of chitin, salicylic acid, auxin, and JA signaling pathways in response to AB infection in a resistant lentil cultivar at the transcriptomic level using the massive analysis of cDNA ends (MACE) technique in AB-resistant and AB-susceptible lentil genotypes.

With the aim to study role of chickpea NBS-LRR resistance genes in AB resistance, expression of 121 NBS-LRR genes distributed across the whole genome was examined in resistant CDC Corinne and CDC Luna genotypes and one susceptible ICC96029 genotype at different time point subjecting AB infection [75]. Five NBS-LRR genes exhibited genotype-specific expression, with *LOC101509145* and *LOC101498915* upregulated in CDC Corinne at 48 and 72 h post-infection (hpi) and downregulated or not regulated in ICCV 96,029 and CDC Luna at all time points and *LOC101512894*, *LOC101513745*, and *LOC101497042* upregulated in ICCV 96,029 and CDC Luna and downregulated or not regulated in CDC Corinne [75]. In addition to RNA-seq, candidate genes situated in AB-resistant QTL regions were also studied for their role in AB resistance. Of the four candidate genes identified as underlying in *qABR4.1* (AB-resistant QTL region) in chickpea [142], the transcripts of one gene, *CaAHL18* gene (belonging to ‘AT-hook motif containing nuclear localized (AHL)’) were induced at higher levels in AB-resistant chickpea accession at 12 hpi and 72 hpi [142]. To obtain insights into the AB infection and AB-resistance mechanisms in chickpea, transcriptome, small RNA sequencing, and degradome sequencing of two AB-resistant and two AB-susceptible chickpea genotypes under control and stress conditions was performed [76]. Garg et al. [76] undertook transcriptome, small RNA, and degradome sequencing of two AB-resistant and two AB-susceptible chickpea genotypes exposed to AB infection, uncovering 6767 DEGs ranging from pathogenesis-related protein encoding genes, NBS-LRR genes, and cell wall biosynthesis genes. Small RNA sequencing identified 297 differentially expressed miRNAs (e.g., nov_miR3a, nov_miR64, nov_miR171, miR3627b, miR2111l, miR2111-3p) involved in controlling AB-resistance in chickpea [76]. Degradome sequencing identified the target genes of these miRNAs [NBS-LRR (*Ca_08122*), Dof zinc finger (*Ca_19433*), and ERF (*Ca_00359*)] contributing to AB resistance [76]. Functional genomics approaches, such as RNA-seq and MACE, have provided novel insights into AB-pathogenicity factors that enable host intrusion/infection in chickpea [15]. Following AB-pathogen invasion, plants generate reactive oxygen species to damage the pathogen propagules. For successful invasion, AB pathogens must overcome oxidative stress generated by the host [198]. Chickpea blight pathogen possess genes to overcome host-induced oxidative stress-mediated injury following host infection [198]. In a nutshell, functional genomics has offered significant advances in discovering candidate gene(s) for AB resistance in legume crops with their possible function in host defense and a few pathogen genes with a putative role in pathogenicity and in overcoming host defenses [75,76].

## 10. Proteomics Approach for Uncovering Key Proteins Contributing to AB Resistance

The proteomics approach is one of the powerful ‘omics’ approaches for increasing our understanding of various proteins and post translational modifications of proteins participating in conferring plant immune responses and disease resistance mechanisms [199]. Hence, quantifying the proteins that render AB resistance could be important for identifying AB-resistant lines. To gain better understanding of the host defense against pathogen attack, Castillejo et al. [200] used shotgun proteomics and data-independent acquisition analysis to identify 83 proteins responding to AB infection in P665, a resistant pea genotype. Of these identified proteins, caffeic acid O-methyltransferase (participates in lignin synthesis and thus strengthens cell walls in response to pathogen attack), 14-3-3-like protein (involved in binding pathogen effectors), and TIC110 defense protein play major roles in mediating AB resistance in pea [200]. 

## 11. Host Plant Legume Genome Sequences and AB Pathogen Genome Sequence: Exploring Host–Pathogen Co-Evolution and Understanding AB Resistance 

Advances in next-generation-based genome sequencing technologies have led to the elucidation of complete genome sequence information for chickpea and pea, enabling the exploration of key genomic regions conferring AB resistance [201,202]. Likewise, genome assemblies of AB-causing pathogens [*A. rabiei* (ITCC No. 4638) [16] and *A. lentis* for the Australian isolate Al4 [203] have been constructed to obtain novel insights into pathogen effector encoding gene(s), pathogenicity gene(s), carbohydrate active enzymes, and secondary metabolite synthesis gene(s). Comparative genomic analysis of the genome sequences of AB-causing pathogen could reveal genome synteny or conserved regions among these species. Lee et al. [203] reported the presence of highly conserved synteny of genomic regions along with several chromosomal rearrangements between *A. lentis* and *A. rabiei* genomes. The authors also uncovered genome homology between *A. lentis* and *A. rabiei* for secondary metabolite gene clusters and effector genes with 40% amino acid similarity in proteins of these genes [203]. More affordable genome sequencing has offered greater opportunities to perform WGRS for elucidating genomic regions conferring disease resistance and other traits across the whole genome in large sets of global germplasm in various legume crops [202,204,205]. Resequencing of 429 global chickpea germplasm [204] and sequencing of 3366 chickpea germplasm [205] uncovered untapped useful allelic variations for various traits of agronomic importance, including various biotic stresses in chickpea. WGRS of 69 chickpea genotypes revealed 12 candidate genes on AB4.1 QTL, encoding *NBS-LRR receptor-like kinase*, *wall-associated kinase*, and *zinc finger protein* [159]. Similarly, pangenome assembly of host plant [205] and AB-causing fungus [206] could lead to identify host plant structural variants (that exist in the accessory genome) contributing towards disease resistance and pathogenesis gene/effector gene(s) of causative fungal pathogen [205,206]. Thus, pangenome of both the legume crops and AB-causing pathogens will provide great insights in the AB-resistance genes in host and as the pathogenesis genes in pathogens for designing AB-resistant grain legumes.

## 12. Phenomics: High-Throughput Phenotyping Approach for Capturing Plant and AB Disease Interaction Dynamics at the Multidimensional Level 

Despite unprecedented advancements in genome sequencing technologies especially next generation sequencing coupled with bioinformatics technologies that facilitate access to gene sequences and gene functions; the information on mechanisms of pathogenicity and host resistance are far from understood. Phenotype information can fill this gap in our understanding of host pathogen interactions and bases of host resistance. Capturing phenotypic information at a large scale still remains a daunting task for crop breeding studies.

The traditional phenotyping approach for quantifying disease reactions of host plants including symptoms and biochemical changes is labor intensive, costly, and time-consuming, thus, limiting precise phenotyping for disease resistance. However, plant phenomics have evolved due to the recent advent of sophisticated sensor-based technologies, advanced imaging technologies, unmanned aerial vehicles (UAV) equipped with advanced sensors, artificial intelligence, and other advanced phenotyping platforms, enabling high-capacity computing to measure plant phenotyping data in multidimensions and at multiscales [207,208,209,210]. These emerging automated platforms have alleviated the ‘micro-phenotyping’ bottleneck and facilitated the capture of host plant and disease reactions at spatial and temporal levels with higher precision [207,210,211]. For example, UAV has been used to monitor disease severity in the field in rice for sheath blight [212], potato for late blight [213], and soybean for powdery mildew [214]. A study phenotyping AB-disease severity in chickpea using an unmanned aircraft system in association with various multispectral cameras captured images of crop canopy area; the vegetation indices revealed significant associations between these images and crop yield and disease severity based on visual ratings [209]. Thus, remote sensing based high-throughput phenotyping can predict AB-disease severity in chickpea and help with the timely application of disease management approaches thereby minimizing yield losses. Likewise, emerging next-generation artificial intelligence, including machine learning, convolutional networks, support vector machines, and deep learning, have been used for field phenomics, including disease detection, and disease symptom characterization [215,216,217,218]. These approaches could facilitate the early detection of various diseases, including AB in grain legumes, for implementing appropriate disease management practices to minimize disease-related losses. 

## 13. Conclusions and Future Prospects

Given global climate change and deployment of resistant host cultivars, plant pathogens including AB pathogens are evolving, leading to new virulences and thus leading to the breakdown of host resistance and increased yield losses in grain legume. To minimize AB-caused yield loss in grain legumes, breeders aim to identify new sources of resistance across various gene pools and transfer those to elite cultivars with the objective of having an economical, sustainable, and environment friendly disease management approach rather than that based on the use of fungicides. Since, resistance to AB is partial and rare in cultigens, CWRs are important sources of AB-resistant gene(s) in breeding programmes. Current advances in genome assembly of various legumes including CWRs and AB pathogens have provided opportunities to identify AB resistance and pathogenicity gene(s) to gain better understanding of host pathogen interactions, isolate resistance genes and develop AB-resistant grain legumes. Likewise, WGRS and pangenome approaches could harness novel structural genomic variants and R gene(s) contributing to AB resistance across the whole-genome/species level in host plants [159] and key information on pathogen effector encoding genomic regions or pathogenicity gene(s) in the AB pathogen. Likewise, rapid advances in functional genomics, especially RNA-seq, has facilitated the discovery of AB-resistant candidate gene(s) and their functions and enriched our understanding of the complex molecular mechanisms of host plant interactions, disease development, and host plant resistance mechanisms. Proteomics offers insight into various host proteins contributing to mediating AB resistance and AB-pathogen toxins responsible for disease development. Phenomics facilitates scoring of diseases at a large scale. To score disease in a large area, quantify AB-disease reaction at multidimensional levels, forecast onset of epidemics and minimize yield losses, high throughput phenotyping approaches including sensor-based technologies and UAVs have revolutionized disease phenotyping including AB under field conditions. The next challenge is integrating the large-scale genomic data obtained through next generation sequencing and phenotypic data obtained through phenomics approaches. Powerful next-generation AI could be used to integrate these ‘big data’ to accelerate the development of climate-resilient crop cultivars [218].

Emerging novel breeding technologies, viz., marker assisted selection, genomic selection, and speed breeding, could be used to select superior recombinants/progenies with high breeding value and AB resistance. Notable instances of these novel techniques have been reported in chickpea [219,220,221] and pea [186] for developing AB-resistant cultivars. However, development of AB resistance employing these novel techniques in other legumes needs more attention. Similarly, CRISPR/Cas9-based genome editing tools that can mutate AB-susceptibility gene(s) in high yielding but AB-sensitive genotypes is another approach to develop AB-resistant grain legume cultivars to ensure global food security.

## Figures and Tables

**Figure 1 ijms-23-02217-f001:**
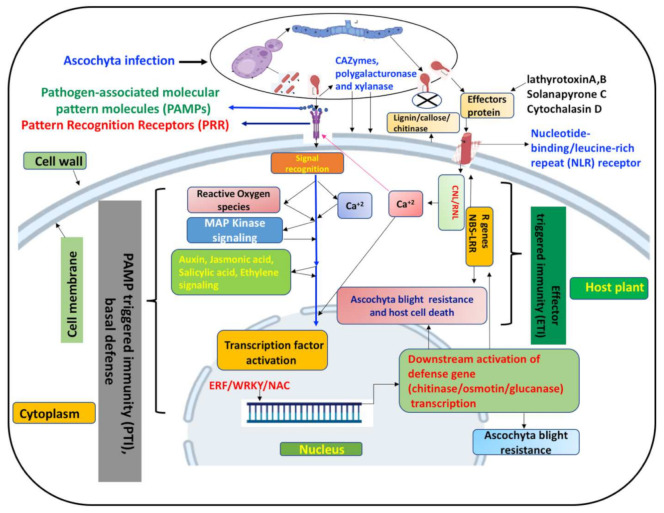
Proposed mechanism of Ascochyta blight (AB) resistance in grain legumes.

**Table 1 ijms-23-02217-t001:** Races/pathotypes of Ascochyta blight and its major symptoms in major grain legumes.

Crop	Scientific Name of Causal Organism	Races/Pathotype	Symptoms
Chickpea	*Ascochyta rabiei* (anamorph); *Didymella rabiei* (teleomorph)	Three pathotypes [29,30], four pathotypes [32], five pathotypes [6], ten pathotypes [33]	Concentric necrotic lesions on all aboveground plant parts [34]; 100% chickpea yield loss under favorable environment [11].
Faba bean	*Ascochyta fabae*	Races 1, 2, 3, and 4 [7]; Seven races [35]	Stem lesions usually darker than leaf and pod lesions; lesions can be produced over the surface; mycelial invasion causes seed infection [36].
Lentil	*Ascochyta lentis* Vassiljevsky (teleomorph: Didymella lentis, syn. *Ascochyta fabae* f. sp. *lentis*) [37]	Two mating types [38]	Symptom appears as necrotic lesions on leaves, stems, and pods, inhibiting photosynthesis and causing up to 70% seed yield losses [5].
Lathyrus	*Ascochyta lentis var. lathyri*	–	Symptoms are characterised by presence of necrotic lesions on stems and leaves [39].
Pea	*Didymella pinodes*, *Phoma pinodella*, and *P. koolunga*, *Ascochyta pisi* Lib., *Mycosphaerella pinodes*, *Phoma medicaginis* var. *pinodella* [8,40,41,42,43,44]	–	Small ‘pinprick’ lesions, flecks on the leaf surface, coalescence of expanding lesions under wet condition, senescence of leaves [45,46] symptoms are noticed. Stem lesions have similar color and elongation, lesions become progressively longer and often coalesce to completely girdle stems [47,48].

**Table 2 ijms-23-02217-t002:** Sources of Ascochyta blight resistance in various grain legumes.

Crop	Name of Accession	Reporting Country	Reference
Chickpea	*C. judaicum*, *C. pinnatifidum*	-	[106]
Chickpea	*C. echinospermum*, *C. reticulatum*	-	[99]
Chickpea	HOO-108, GL92024	India	[107]
Chickpea	PI 559361, PI 559363, W6 22589	USA	[108]
Chickpea	RIL58-ILC72/Cr5	Spain	[109]
Chickpea	Almaz, ICC 3996, ILWC 118	Australia	[88]
Chickpea	FLIP 98-133C, FLIP 98-136C	Canada	[110]
Chickpea	FLIP 97-121C	India	[111]
Chickpea	FLIP 4107, FLIP 1025, FLIP 10511	Algeria	[104]
Chickpea	EC 516934, ICCV 04537, ICCV 98818, EC 516850, EC 516971	India	[103]
Chickpea	ICC7052, ICC4463, ICC4363, ICC2884, ICC7150, ICC15294, ICC11627	Kenya	[102]
Chickpea	10A, 28B	Turkey	[112]
Chickpea	ILC72, ILC182, ILC187, ILC200, ILC202	Algeria	[113]
Chickpea	*C. echinospermum* accessions S2Drd_ 061 Deste_064, *C. reticulatum* accession Bari1_062, *C. echinospermum* accession Karab_063		[114]
Chickpea	*Cicer echinospernum*	NSW-DPI, Tamworth, Australia	[115]
Chickpea	IC275447, IC117744, EC267301, IC248147, EC220109	PAU, Ludhiana, India; HAREC, HPKV, Dhaulakuan India	[105]
Faba bean	BPL 471, 2485	Syria, England, Canada, Poland, France, Tunisia	[7]
Faba bean	SU-R 40, SU-R 5/13	-	[116]
Faba bean	ILB 1414, ILB 6561	-	[98]
Faba bean	Ascot	-	[117]
Faba bean	29H	France	[93,94]
Faba bean	ILB 752	-	[92]
Faba bean	V-1220, V-494, V-175, V-47, V-165, V-1122, V-46	-	[118]
Faba bean	L-831818, V-26, V-958, V-255, V-1020, V-1085, V-1117, V-1020, V-1085, V-1117	Czech Republic, Estonia, Germany, Spain	[58]
Grasspea	*L. sativus*, *L. ochrus*, *L. clymenum*	-	[119]
Lentil	Indian head	-	[89]
Lentil	W6 3261, W6 3192, W6 3222, W6 3241	-	[23,26]
Lentil	24 *Lentis orientalis*, 12 *Lentis odemensis*, 3 *Lentis nigricans*	-	[99]
Lentil	ILL 358, ILL 5588, ILL 5684, Laird	Canada	[68]
Lentil	ILL 358, ILL 4605	Chile	[68]
Lentil	ILL 358, LL 857	Ethiopia	[68]
Lentil	ILL 5698, ILL 5700, ILL 5883, ILL 6212	Morocco	[68]
Lentil	ILL 5684, ILL 5588, ILL 5714, Rajah	New Zealand	[68]
lentil	FLIP84-27L, FLIP84-43L, FLIP84-55L	Pakistan	[68]
Lentil	ILL 857, ILL 2439, ILL 4605	Syria	[68]
Lentil	HPL5, L442, L448, LG169, Pant4, Pant L406	India	[68]
Lentil	ILL 7537	Jordan	[120]
Lentil	*L*. *ervoides*, *L. nigricans*, *L. culinaris* subsp. *orientalis*, *L. culinaris* subsp. *odemensis*	Saskatoon, Canada	[121]
Lentil	ILL 1704	Ethiopia	[121]
Lentil	CDC Robin, 964a-46	-	[122]
Lentil	*L. orientalis* accession ILWL 180	ICARDA	[123]
Lentil	IG207	ICARDA	[124]
Lentil	ILL7537	-	[70]
Pea	Lines JI 96, JI 252, JI 1089	Afghanistan, Ethiopia, Syria	[125]
Pea	*P. fulvum* accession PS1115	-	[119]
Pea	*P. fuivum*, followed by *P. sativum* ssp. *eiatius* and *P. sativum* ssp. *syriacum*	-	[100]
Pea	*Pisum fulvum* accessions PI595937, P651, W615017, PI560061, *P*. *sativum* subsp. *elatius* accession PI344538	Canada	[127]
Pea	P13 *Pisum sativum* ssp. *elatius*	Turkey	[14]
Pea	P18 *Pisum sativum* ssp. *elatius*	Greece	[14]
Pea	P651 *Pisum fulvum*	Syria	[14]
Pea	P665 *Pisum sativum* ssp. *syriacum*	Syria	[14]
Pea	P670 *Pisum sativum* ssp. *elatius*	Turkey	[14]
Pea	05P778-BSR-701, ATC 5338, ATC 5345, Dundale, ATC 866	Western Australia	[48]
Pea	*Pisum fulvum* accessions PI595937, P651, W615017, PI560061, *P. sativum* subsp. *elatius* accession PI344538	Canada	[128]

**Table 3 ijms-23-02217-t003:** List of Aschochyta blight resistant QTLs identified in various grain legumes.

Crop	Mapping Approach	Mapping Population	QTL/Gene	Type of Marker Used	Linkage Group (LG)	Phenotypic Variation (PV) %	Reference
Chickpea	Bi-parental	FLIP84-92C × C. *reticulatum* Lad. (PI 599072)	QTL-1 and QTL-2	RAPD, ISSR	-	50.30	[147]
Chickpea	Bi-parental	Lasseter × C. *echinospermum* accession (PI 527930), F2	1 QTL	STMS	LG4	-	[144]
Chickpea	Bi-parental	ILC 1272 × ILC 3279	*ar1*, *ar2a*, *ar2b*	SSR	LG4, LG2	-	[145]
Chickpea	Bi-parental	PI 359075 × FLIP84-92C F_7_(RIL)	3 QTL + Ar19 (or Ar21d) gene	SSR	(LG)4A, LG2 + 6	-	[148]
Chickpea	Bi-parental	*Cicer arietinum* (ILC72) × *Cicer reticulatum* (Cr5-10)	1 QTL	RAPD, ISSR, STMS, isozyme	LG2	28	[140]
Chickpea	Bi-parental		QTL_AR3_	STMS	LG2	-	[149]
Chickpea	Bi-parental	ICCV96029′ and ‘CDC Frontier (186 F_2_)	T3hree QTL	SSR	LG3, 4, 6	12–29	[150]
Chickpea	Bi-parental	CDC Frontier × ICCV 96029, CDC Luna × ICCV 96029, CDC Corinne × ICCV 96029, Amit × ICCV 96029, F_1_ and F_2_	5 QTL (QTL1–5)	SSR	LG 2, 3, 4, 6, 8	14–56	[151]
Chickpea	Bi-parental	ICC 4991 × ICCV 04516	3 QTL	SSR	LG3, LG4	7.7–18.6	[152]
Chickpea	Bi-parental	*C. arietinum* × *C. reticulatum*	3 QTL	SSR	LG3, LG4	49	[153]
Chickpea	Bi-parental	ILC3279 × WR315, RIL	QTL(AR1), EIN4-like sequence	SSR	LG4	33.8	[139]
Chickpea	Backcross	CDCXena × CDC Frontier, CDCXena × CDC 425-14	*Abr QTL 3*, *Abr QTL 4*	SSR	LG4, LG8	-	[154]
Chickpea	Bi-parental	C 214′ × ‘ILC 3279′ (F2)	*AB*-*Q*-*SR*-*4*-*1*, *AB*-*Q*-*SR*-*4*-*2*, *AB*-*Q*-*APR*-*6*-*1*, *AB*-*Q*-*APR*-*6*-*2*, *AB*-*Q*-*APR*-*4*-*1*, *AB*-*Q*-*APR*-*5B*	SSR	LG4, 5, 6	1.5–32	[155]
Chickpea	-	-	42 candidate genes *Ein3*, *Avr9/Cf9 and Argonaute 4*	SNP	Ca2	44.2	[156]
Chickpea	Bi-parental	S95362 × Howzat, Lasseter × ICC3996	*ab_QTL1*, *ab_QTL2*	EST, SNP	-	14–45	[157]
Chickpea	Bi-parental	ICCV 96029 × CDC Frontier RIL(92)	*qtlAb-1.1*, *qtlAb-2.1*, *qtlAb-3.1*, *qtlAb-4.1*, *qtlAb-6.1*, *qtlAb-7*, *qtlAb-8.1*, *qtlAb-8.2*, *qtlAb-8.3*	SNP	LG1, 2, 3, 4, 6, 7, 8	9–19	[158]
Chickpea	GWAS	132 advanced breeding lines	*AB4.1* QTL along with 12 candidate genes	SNP	LG4	-	[159]
Chickpea	Bi-parental	JG 62 × ICCV 05530, RIL(188)	1 QTL for seedling resistance; minor QTL each for SR and adult plant resistance	SSR, SNP	-	-	[160]
Chickpea	Bi-parental	FLIP84-92C(2) × PI359075(250 RILs), CRIL-7 (217 RILs)	*qABR4.1*, *qABR4.2*, *qABR4.3* QTL and *CaAHL18* candidate gene	SNP	LG4	42	[142]
Chickpea	GWAS	146 (C. *reticulatum*) + 44 (C. *echinospermum*)	WRKY TF *(Cr_02657.1)*, *(Cr_09847.1)* encodes a TF of ARF family	SNP	LG3, 4, 6	6.7–15.2	[114]
Chickpea	Bi-parental	ICCV 96029 × CDCFrontier 92 RILs, ICCV 96029 × Amit 139 RILs	*CPR01-qAB1.1*, *CPR01-qAB1.2*, *CPR01-qAB1.3*, *CPR01-qAB1.4*, *CPR01-qAB4.1*, *CPR01-qAB4.2*, *CPR01-qAB4.3*, *CPR01-qAB4.4*, *CPR01-qAB4.5*, *CPR01-qAB6.1*, *CPR01-qAB6.2*, *CPR01-qAB7.1*	SNP	LG1, 2, 3, 4, 6, 7, 8	-	[161]
Chickpea	Bi-parental	Amit × ICCV 96029 (RIL)	8 QTL	SNP	LG2, 3, 4, 5 and 6	7–40	[162]
Chickpea	Bi-parental	*C. arietinum* × *Cicer echinospernum*, RIL(134)	*AB_echino_2014*, *AB_echino_2015*	SNP	LG4	34–41	[138]
Chickpea	Bi-parental	GPF2 × *C. reticulatum* acc. ILWC 292, 187 RIL	*qab-4.1*, *qab-4.2*, *qab-7.1*	SNP	LG4, LG7	7–11	[163]
Faba bean	Bi-parental	196F_2_	*Af1* and *Af2*	-	LGVIII, LGIVa	46	[164]
Faba bean	Bi-parental	29H × Vf136, (F_2_)	6 QTL (*Af3* to *Af8*)		LG2, LG3, LG6, LG12, LG14, LG15	6.2–44.7	[165]
Faba bean	Bi-parental	Vf6 Vf136 (165 RIL)	*Af1* and *Af 2*	RAPD, SSR, Isozyme EST, SCAR	LG2, LG3	16–24	[166]
Faba bean	Bi-parental	Icarus × Ascot	QTL-3, QTL-1, QTL-2, QTL-4	SSR, SNP	Chr-II, Chr-VI, Chr-I.A		[167]
Faba bean	Bi-parental	29H × Vf136, (RIL)	10 QTL	-	LG2 LG3, LG4	9.8–17.7	[57]
Faba bean	Bi-parental	29H × Vf136 (RIL, 119)	9 QTL	SNP	chromosomes II, III, IV	10.6–21.4	[56]
Faba bean	Bi-parental	29H × Vf136, Vf6 × Vf136	*Af2*, *Af3*, *F_DSP1*, *F_DSP2 and DSL_Lo98*		Chromosomes II, III, IV	7.8–14	[143]
Faba bean	GWAS	188 diverse lines	12 MTAs	AFLP, SNP	LGI, III, IV, V, VI	5.6–21.7	[168]
Lathyrus	Backcross population	ATC 80,878 × ATC 80407	*QTL1*, *QTL2*	RAPD, STMS, STS/CAPS	LG1, LG2	9–12	[169]
Lentil	Bi-parental	ILL5588 (cv. Northfield) × ILL6002	-	RAPD	-	89	[20]
Lentil	Bi-parental	Eston x Indian head, F_2_(60)	*ral* 2gene	RAPD, SCAR	-	-	[170]
Lentil	Bi-parental	ILL5588 (cv. Northfield) × L692-16-1	2 QTL	RAPD, ISSR, RFLP, AFLP	LG4	36	[171]
Lentil	F2	ILL5588 (cv. Northfield) × ILL7537, ILL7537 × ILL6002	5 + 3 QTL	RAPD, ISSR, AFLP	LG1, 2, 4, and 5	50	[120]
Lentil	Bi-parental	Eston × PI 320,937	1 QTL	RAPD, AFLP, SSR	LG6	41	[172]
Lentil	Bi-parental	ILL5588 (cv. Northfield) × ILL5722 (cv. Digger) F_5_(94)	QTL1, QTL2, QTL3, QTL4, QTL5, QTL6	ST-SSR/SSR, ISSR, RAPD, ITAP	LG1, 4, 5 and 9	34–61	[173]
Lentil	Bi-parental	Indianhead × Northfield	*AB_IH1*, *AB_IH1.2*, *AB_NF1*, *AB_IH1.3*	SSR, SNP	LG2, 3 and 6	7–47	[138]
Lentil	Bi-parental	*Lens culinaris* × *L*. *odemensis*, RIL	AS-Q1, AS-Q2, AS-Q3	SNP	LG6	23–27	[133]
Pea	Bi-parental	3148-A88 × Rovar, F_2:4_	13 QTL	RAPD, STS			[174]
Pea	Bi-parental	Carneval × MP1401	3 QTL	AFLP, SCAR	LG2, 3, 4, 5, 7	36	[154]
Pea	Bi-parental	P665 × Messire	6 QTL	RAPD, STS, EST	LG2, 3, 4, 5	31–75	[175]
Pea	Bi-parental	DP × JI296 (135 RIL)	6 QTL at the seedling stage	-	-	56.6–74	[137]
Pea	Bi-parental	A26 × Rovar, A88 × Rovar	11 + 14 QTL	STS	LG I, II, III, IV, V, VI, VII	4.6–37.4	[176]
Pea	Bi-parental	JI296 × DP RIL	RGA-G3A, RGA2.97, PsPRP4A, Peachi21, PsMnSOD, DRR230-b, PsDof1, peabetaglu and DRR49a, *QTL mpIII-4*	-	LG2, 3, 7, 4, 6	-	[177]
Pea	Bi-parental	P665 × Messire	3 QTL	SSR	-	-	[178]
Pea	Bi-parental	A26 × Rovar, A88 × Rovar.	*Asc2.1*, *Asc4.2*, *Asc4.3* and Asc7.1 QTL, 14 candidate genes	–	-	-	[179]
Pea	Bi-parental	P651 (*P*. *fulvum*) × Alfetta (*Pisum sativum* L.) RIL(144)	*abI-IV-1*, *abI-IV-2*, *abI-IV-3*, *abI-IV-4*, *abIII- 1*, *abVII-1*, *abI-IV-5*, *abIII-2*, *abVII-2*	SNP	LG1, 2, 3, 4, 7	7.5–28%	[128]
Pea	Bi-parental	F_6_ RILs PR-19-224 and PR-19-173	*abI-IV-2.1* and *abI-IV-2.2*	SNP	-	5.5–14%	[141]
Pea	Genome-wide association study	36 cultivars	3 MTAs	SNP	-	-	[180]

**Table 4 ijms-23-02217-t004:** Differentially expressed genes (DEGs)/candidate genes associated with ascochyta blight resistance in grain legumes along with putative function.

Crop	DEG/Candidate Gene	Function	References	Genotype Name	Technique Used
Chickpea	97 DEGs	Pathogenesis-related proteins, proline-rich protein, SNAKIN2 antimicrobial peptide, leucine-zipper protein	[193]	ICC3996, FLIP94-508C, ILWC245	RT-PCR, Microarray technology
Chickpea	*LOC101508336*, *LOC101508648*, *LOC101508966*, *LOC101509280*	–	[142]	FLIP8492C, PI359075	qRT-PCR
Chickpea	6767 differentially expressed genes, 651 miRNAs, chitinases (*Ca_04405*), CC-NBS-LRR (*Ca_08361*), CC-NBS-LRR (*Ca_08122*), Dof zinc finger protein (*Ca_19433*), ERF (*Ca_00359*), calcium-transporting ATPase (*Ca_12185*), senescence-associated protein (*Ca_15107*), cellulose synthase (*Ca_08607*)	Pathogenesis-related proteins, cell wall synthesis, NBS-LRR, secondary metabolites	[76]	ILC 3279, ICCV 05530, C 214, Pb 7	Illumina Inc., San Diego, CA, USA, qRT-PCR
Faba bean	850 differentially expressed transcript	Biosynthesis of secondary metabolites, ethylene, phenylpropanoid and isoflavonoids, NBS-LRR proteins synthesis	[194]	29H and Vf136	Illumina platform, RT-qPCR
Lathyrus	29 unique gene sequences	Pathogen recognition, signaling transduction, transcription regulation, PR proteins, and disease resistance	[192]	ATC 80,878, ATC 80407	Microarray technology
Lentil		Pathogenesis-related proteins, genes related to hormone signaling, cell death, and cell-wall reinforcement	[122]	CDC Robin, 964a-46 Eston	HiSeq 2500. qRT-PCR
Lentil	pathogen invasion recognition and signaling genes, pathogenesis-related protein genes, ethylene response factor (ERF)	Fungal elicitors recognition, defense signaling genes, hypersensitive reaction and cell death, transcription regulation of defense genes	[70]	‘ILL7537′ and ‘ILL6002′	RT- qPCR. RNA-Seq
Lentil	Lignin biosynthesis, jasmonic acid pathway signaling gene	Contributed to defense response	[195]	Lupa, ILL5588, BG16880	Massive analysis of cDNA ends
Pea	346 DEGs	Pathogenesis-related (PR) proteins, hormone signaling, cell wall reinforcement, phenylpropanoid	[196]	P665	Microarray technology

## Data Availability

The data presented in this study are available in the article.

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
