# Peer review of "Breeding and Genomics Interventions for Developing Ascochyta Blight Resistant Grain Legumes"

_ijms, 2022, doi:10.3390/ijms23042217_

Round 1
Reviewer 1 Report
The review entitled "Breeding and genomics interventions for developing Ascochyta 2
blight-resistant grain legumes" is a comprehensive review with a good compilation of data.
However, it needs a lot of minor changes before acceptance. Authors should read it carefully and revise it with dedication.
Few comments are below
Q1. Figure 1 should be redrawn with good representation and more clarity.
Q2. Authors should tabulate various information which is already in the text.
Q3. Authors should put a main subheading 2. "Causal organism of AB in legume, symptoms, and negative impact"
then subheading 2.1 chickpea
2.2 faba bean
2.3 pea and so on..
Q4. Avoid the use of unwanted abbreviations such carbohydrate active enzymes (CAZymes) 47;
as next-generation sequencing(NGS) at line 60;
high-throughput phenotyping (HTP) at line 69;
pattern recognition receptors (PRRs) line 139; BAK1 in 142;
focused identification of germplasm strategy(FIGS) at line 261; TF etc
remove all abbreviations, which is hardly used 2-3 times, its can creat confusion for readers.
Q5. "7. Aschochyta blight infection and underlying host plant resistance mechanism" start with subheading 3.
Q6. Rewrite section "16. Proteomics approach for uncovering key proteins contributing to AB resistance". discuss only the Proteomics approach.
Q7. At a various place, Authors used both formats such as Maurin
et al. (1993)[91] in line 212 and Timmerman-Vaughan et al. (2002)[177] in 399 etc. use proper style.
Author Response
Response reviewer 1
Q1. Figure 1 should be redrawn with good representation and more clarity.
- Response: we have redrawn the Fig1with more clarity indicating the AB infection process with details.
Q2. Authors should tabulate various information which is already in the text.
Response: we have tabulated various AB resistant genotypes in each legume in Table 2 and list of QTLs controlling AB resistance in legumes with details in Table 3 and various gene(s)/differentially expressed genes for AB resistance in Table 4.
Q3. Authors should put a main subheading 2. "Causal organism of AB in legume, symptoms, and negative impact"
then subheading 2.1 chickpea
2.2 faba bean
2.3 pea and so on..
- Response: We have done the changes and highlighted in yellow in the text.
Q4. Avoid the use of unwanted abbreviations such carbohydrate active enzymes (CAZymes) 47;
as next-generation sequencing(NGS) at line 60;
high-throughput phenotyping (HTP) at line 69;
pattern recognition receptors (PRRs) line 139; BAK1 in 142;
focused identification of germplasm strategy(FIGS) at line 261; TF etc
remove all abbreviations, which is hardly used 2-3 times, its can creat confusion for readers.
- Response: We have deleted all the above-mentioned abbreviations highlighted in pink in the text.
Q5. "7. Aschochyta blight infection and underlying host plant resistance mechanism" start with subheading 3.
- Response: We have included the subheadings as suggested by the reviewer
Q6. Rewrite section "16. Proteomics approach for uncovering key proteins contributing to AB resistance". discuss only the Proteomics approach.
- Response: We have rewritten the proteomics section highlighted in pink in the text.
Q7. At a various place, Authors used both formats such as Maurin
et al. (1993)[91] in line 212 and Timmerman-Vaughan et al. (2002)[177] in 399 etc. use proper style.
- Response: We have used proper style for the above references in the text.

Reviewer 2 Report
The manuscript reviews the results of previous studies investigating the resistance of legumes to diseases caused by Didymella spp. Legumes are an important source of dietary protein in many regions of the world. The manuscript is extensive, and the findings can be of interest to specialists dealing with resistance breeding. However, since the manuscript covers a wide range of topics, some issues were only superficially addressed. The sections describing the sources of resistance and identification of Ascochyta blight - resistant quantitative trait loci (QTL) using biparental mapping and functional genomics are relatively detailed and particularly valuable.
Unfortunately, as many as 41 references (especially those cited in the first section) were published in the 20th century, whereas only 13 papers were published in the last three years. Therefore, some information appears to be outdated and should be supplemented.
- 41: replace “(imperfect stage: Ascochyta)” with “anamorph: Ascochyta”
In the descriptions of pathogens and diseases, teleomorph names should be provided.
- 74: I would prefer: “Didymella rabiei (Kovatsch.) Arx. (anamorph: Ascochyta rabiei (Pass.) Labr)”.
Teleomorph names are given in recent studies.
Sanae Krimi Bencheqroun et al Pathogen diversity and mating types of Didymella rabiei isolates collected from Morocco, Current Plant Biology, Volume 29, 2022, 100231, ISSN 2214-6628, https://doi.org/10.1016/j.cpb.2021.100231.
- 101. The Authors should take into account changes in the taxonomy of species infecting pea, which are presented, among others, in the below publications:
Barilli E, et al. Clarification on Host Range of Didymella pinodes the Causal Agent of Pea Ascochyta Blight. Front Plant Sci. 2016 May 13;7:592. doi: 10.3389/fpls.2016.00592. PMID: 27242812; PMCID: PMC4865514.
Šišić A. et al. Novel Real Time PCR Method for the Detection and Quantification of Didymella pinodella in Symptomatic and Asymptomatic Plant Hosts. Journal of Fungi. 2022; 8(1):41. https://doi.org/10.3390/jof8010041
- 119. The description should be expanded, this section consists of only one sentence (?); please note that all sections describing diseases could be merged into one section.
Marco Masi et al. 2018. Lathyroxins A and B, Phytotoxic Monosubstituted Phenols Isolated from Ascochyta lentis var. lathyri, a Fungal Pathogen of Grass Pea (Lathyrus sativus). J. Nat. Prod. 2018, 81, 4, 1093–1097. Publication Date: February 28, 2018. https://doi.org/10.1021/acs.jnatprod.7b01034
L: 224-233. This section is not related to the topic and could be deleted.
L: 3019: The citation is incorrect.
L: 439-452. This section is very general and could be deleted.
The conclusion simply summarizes all sections of the manuscript, whereas new research directions should be emphasized. Since all legume species are described in sections 8-18, the Authors could indicate the species characterized by the fastest breeding progress.
Author Response
Response reviewer 2
Unfortunately, as many as 41 references (especially those cited in the first section) were published in the 20th century, whereas only 13 papers were published in the last three years. Therefore, some information appears to be outdated and should be supplemented.
- Response: We thank the reviewer for the comments. We have minimised the number of old old references; however, some seminal papers were kept because these are very important studies. We have also updated new references highlighted in yellow inreference section.
- 41: replace “(imperfect stage: Ascochyta)” with “anamorph: Ascochyta”
- Response: We have replaced it and highlighted in yellow
In the descriptions of pathogens and diseases, teleomorph names should be provided.
- 74: I would prefer: “Didymella rabiei (Kovatsch.) Arx. (anamorph: Ascochyta rabiei (Pass.) Labr)”.
- Response: We have referred as “Didymella rabiei(Kovatsch.) Arx.
Teleomorph names are given in recent studies.
Sanae Krimi Bencheqroun et al Pathogen diversity and mating types of Didymella rabiei isolates collected from Morocco, Current Plant Biology, Volume 29, 2022, 100231, ISSN 2214-6628, https://doi.org/10.1016/j.cpb.2021.100231.
- Response: We have added this reference highlighted in yellow in both text and in reference section
- 101. The Authors should take into account changes in the taxonomy of species infecting pea, which are presented, among others, in the below publications:
Barilli E, et al. Clarification on Host Range of Didymella pinodes the Causal Agent of Pea Ascochyta Blight. Front Plant Sci. 2016 May 13;7:592. doi: 10.3389/fpls.2016.00592. PMID: 27242812; PMCID: PMC4865514.
Šišić A. et al. Novel Real Time PCR Method for the Detection and Quantification of Didymella pinodella in Symptomatic and Asymptomatic Plant Hosts. Journal of Fungi. 2022; 8(1):41. https://doi.org/10.3390/jof8010041
- Response: We have added these reference highlighted in yellow in both text and in the reference section.
- 119. The description should be expanded, this section consists of only one sentence (?); please note that all sections describing diseases could be merged into one section.
- Response: We have merged the disease section with broad heading 2 and its sub section 2.1,2.2,2.3,2.4 and 2.5 are mentioned in the text and highlighted in yellow .
Marco Masi et al. 2018. Lathyroxins A and B, Phytotoxic Monosubstituted Phenols Isolated from Ascochyta lentis var. lathyri, a Fungal Pathogen of Grass Pea (Lathyrus sativus). J. Nat. Prod.2018, 81, 4, 1093–1097. Publication Date: February 28, 2018. https://doi.org/10.1021/acs.jnatprod.7b01034
- Response: We have corrected it in the reference section highlighted in yellow.
L: 224-233. This section is not related to the topic and could be deleted.
- Response: We have deleted this section.
L: 3019: The citation is incorrect.
- Response: We have corrected it.
L: 439-452. This section is very general and could be deleted.
- Response: We have deleted it.
The conclusion simply summarizes all sections of the manuscript, whereas new research directions should be emphasized. Since all legume species are described in sections 8-18, the Authors could indicate the species characterized by the fastest breeding progress.
- Response: We have added the references for the legumes having implication of novel emerging breeding techniques for combating AB resistance and mentioned the scope of these techniques for other legumes for AB resistance in the conclusion section.

Reviewer 3 Report
The manuscript deals with an overview of Ascochyta blight, causal agent, its genetic control, resistence in cultivated fabaceae.The authors have shown many recent studies especially of molecular, omics ones and therefore meets the expectations of International Journal of Molecular Sciences.
The manuscript is well structured and easy to read and therefore with high potential to be shown.
The two main concerns are
1-the lack is the absence of economic impact and production decrease around the world. May be, it can be found on the database of FAO?
2- studies done using classical methods for determination of inheritance of resistence to Ascochyta blight.
Item 8. Genetics of AB resistance
See
https://jacobspublishers.com/journals/jacobs-journal-of-agriculture/fulltext/analysis-of-genetic-components-of-resistance-to-ascochyta-rabiei-in-chickpea
Minor remarks
Keywords: please change Ascochyta blight by another key word since it is present in title.
L205 please change sohown by shown
Author Response
Response reviewer 3
The manuscript deals with an overview of Ascochyta blight, causal agent, its genetic control, resistance in cultivated fabaceae.The authors have shown many recent studies especially of molecular, omics ones and therefore meets the expectations of International Journal of Molecular Sciences.
The manuscript is well structured and easy to read and therefore with high potential to be shown.
The two main concerns are
1-the lack is the absence of economic impact and production decrease around the world. May be, it can be found on the database of FAO?
- Response: We have separately mentioned the loss caused by AB in respective crop globally highlighted in green colour please see in each 2.1,2.2,2.3 2.4. and 2.5 headings. However, there is no report of AB caused losses data in legume on FAO website.
2- studies done using classical methods for determination of inheritance of resistence to Ascochyta blight.
Item 8. Genetics of AB resistance
See
https://jacobspublishers.com/journals/jacobs-journal-of-agriculture/fulltext/analysis-of-genetic-components-of-resistance-to-ascochyta-rabiei-in-chickpea
- Response: We have discussed thoroughly the genetics of AB resistance in each legumes in section 4
Minor remarks
Keywords: please change Ascochyta blight by another key word since it is present in title.
- Response: We have used the AB for ascochyta blight throughout the manuscript highlighted in green colour.
L205 please change sohown by shown
- Response : We have changed it
